# Inhibition of Nematocyst Discharge from *Pelagia noctiluca* (Cnidaria: Scyphozoa)—Prevention Measures against Jellyfish Stings

**DOI:** 10.3390/md20090571

**Published:** 2022-09-08

**Authors:** Ainara Ballesteros, Carles Trullas, Eric Jourdan, Josep-Maria Gili

**Affiliations:** 1Department of Marine Biology and Oceanography, ICM-CSIC-Institute of Marine Sciences, Passeig Marítim de la Barceloneta 37-49, 08003 Barcelona, Spain; 2Innovation and Development, ISDIN, C. Provençals 33, 08019 Barcelona, Spain

**Keywords:** cnidarian, cnidocyte, first-aid, inhibitor effect, Mediterranean Sea, nematocyst, pain, sunscreen

## Abstract

*Pelagia noctiluca* stings are common in Mediterranean coastal areas and, although the venom is non-lethal, they are painful. Due to its high toxicity and abundance, *P. noctiluca* is considered a target species for the focus of research on active ingredients to reduce the symptoms of its sting. To determine the effect of 31 substances and formulations on nematocyst discharge, we performed three tests: (1) screening of *per se* discharge activator solutions, (2) inhibitory test with nematocyst chemical stimulation (5% acetic acid) and (3) inhibitory test quantifying the hemolytic area. Ammonia, barium chloride, bleach, scented ammonia, carbonated cola, lemon juice, sodium chloride and papain triggered nematocyst discharge. All of them were ruled out as potential inhibitors. Butylene glycol showed a reduction in nematocyst discharge, while the formulations of 10% lidocaine in ethanol, 1.5% hydroxyacetophenone in distilled water + butylene glycol, and 3% Symsitive^®^ in butylene glycol inhibited nematocyst discharge. These last results were subsequently correlated with a significant decrease in hemolytic area in the venom assays versus seawater, a neutral solution. The presented data represent a first step in research to develop preventive products for jellyfish stings while at the same time attempting to clarify some uncertainties about the role of various topical solutions in *P. noctiluca* first-aid protocols.

## 1. Introduction

Jellyfish are venomous animals belonging to the Phylum Cnidaria [1,2], which is characterized by the unique presence of stinging cells termed cnidocytes [3,4]. Venom, composed of enzymes, potent pore-forming toxins and neurotoxins [2], is stored in a capsule inside the cnidocyte called the cnidocyst, along with a coiled tubule that may be armed with spines [5,6]. Cnidocytes provide a defense mechanism and are used in prey capture [7], and are one of the largest and most complex intracellular secretion products known [3]. Cnidocysts can be classified into nematocyst, spirocyst and ptychocyst categories, the nematocyst category being the most common among cnidarian specimens, with more than 30 morphological types [6,8].

Due to their toxicity, jellyfish stings have a negative impact on human health [9,10]. It is estimated that 150 million people worldwide are exposed to jellyfish annually [11], and encounters with some cubozoan species in tropical and subtropical waters can result in serious health problems (e.g., cardiopulmonary arrest) [12,13]. However, the jellyfish that inhabit the Mediterranean Sea are considered non-life-threatening species [14,15,16]. Although cases of anaphylaxis can occur [17,18], their stings generally involve skin damage and local symptoms, but can be extremely painful [9,19].

The mauve stinger, *Pelagia noctiluca* is a holoplanktonic scyphozoan present in warm and temperate waters [20]. In the Mediterranean Sea, *P. noctiluca* is considered the most important Mediterranean jellyfish [19,21] with predominant occurrence all year-round but with interannual differences in intensity [15]. This scyphozoan is a highly toxic species, and its venom has hemolytic, cytolytic and dermonecrotic properties, among others [19,21]. Due to its wide distribution along the basin [15,22] and high toxicity [19,21], *P. noctiluca* results in a large number of beachgoers seeking assistance from medical and rescue services on the Mediterranean beaches of Spain, Italy, France and Morocco [10,18,22,23,24]. Its cnidome—a term that includes the total complement of cnidocytes within a cnidarian specimen [25]—is composed of four nematocyst types (a-isorhiza, A-isorhiza, O-isorhiza and eurytele), of which the O-isorhiza and eurytele types play an important role in its sting [25]. *P. noctiluca* stings can produce mild to severe skin lesions involving pain, urticaria, edema, burning sensation, and formation of vesicles, papules, and/or scabs [19,21,23,26]. Furthermore, in rare cases, systemic reactions to a sting can occur such as Guillain-Barré syndrome [27].

In the sting management scenario, inhibition of cnidocyst discharge is an important tool to prevent or reduce the effect of a jellyfish sting [9,28,29,30,31]. Inhibitory substances can be incorporated as prophylaxis in sunscreens to prevent stings [11,32,33], and in first-aid products to avoid further cnidocyst discharge [23,28,30]. Although jellyfish stings are common, the scientific community remains in disagreement about the effectiveness of most substances and products [9,13,28,30,31]. The discrepancies and confusion are due to limited data for making evidence-based recommendations and the lack of a regulated and agreed framework among researchers [28,34,35]. Moreover, in most cases, evidence is based on empirical knowledge that is difficult to transfer to industrial processes due to a knowledge gap in the mechanisms of action, difficulty using these compounds in cosmetic products and the scarce effectiveness [34].

For species belonging to the Cubozoa class, the scientific evidence strongly supports the use of vinegar or 5% acetic acid to safely rinse the sting area [28,30,36]. In fact, some products based on acetic acid are marketed for the treatment of cubozoan stings (e.g., Sting No More^®^) [30]. However, vinegar and 5% acetic acid produce cnidocyst discharge in scyphozoans [28,36,37,38,39], including *P. noctiluca* [28], so it is not recommended to use them or products based on them. Under these premises, and with the aim of developing a safe framework for tourists and beach users, it is essential to re-evaluate topical compounds that have traditionally been used in this situation and discover new ones [34]. Therefore, the present research focused on the reassessment of traditional compounds and the proposal of new potential inhibitors of nematocyst discharge in *P. noctiluca*, a jellyfish that is highly toxic, abundant and widely distributed in the Mediterranean Sea.

## 2. Results

### 2.1. Evaluation of Nematocyst Discharge

#### Test 1—Solution Screening

To reassess existing traditional solutions, in addition to proposing new ones, we examined the effect of potential compounds using the tentacle solution assay (TSA). The solutions that induced discharge *per se* were discarded as potential inhibitors in the following assay (Section 2.2.1) (Figure 1B–M).

The incubation of ammonia, barium chloride, bleach, lemon juice and scented ammonia triggered massive discharge of nematocysts (Figure 1B,C,I,K,L and Table 1). Sodium bicarbonate solutions, sodium chloride, papain, acetic acid, carbonated cola and vinegar elicited medium nematocyst discharge (Figure 1D–H,J,M and Table 1). The rest of the compounds, including seawater (Figure 1A), did not trigger nematocyst discharge from the tentacle after their incubation, so they were classified as neutral solutions and considered potential inhibitors (Table 1).

### 2.2. Evaluation of Inhibitory Effect

#### 2.2.1. Test 2—Nematocyst Discharge

To evaluate the inhibitory effect, nematocyst discharge was chemically stimulated by the application of 5% acetic acid (Figure 2A and Table 2). After incubation and following the chemical stimulation, bromelain, choline chloride, copper gluconate, gadolinium, iodine solutions, lanthanum chloride, magnesium chloride, magnesium sulfate, distilled water, fresh water, physiological saline and urine were classified as neutral solutions (not inhibitory) (Table 2). Nematocyst tubules were observed immediately after the 5% acetic acid application. Only isolated nematocysts were discharged in the presence of butylene glycol, so it was classified as a reducer (Figure 2B and Table 2). After the stimulation with 5% acetic acid, the compounds of lidocaine in ethanol, 1.5% hydroxyacetophenone in distilled water + butylene glycol, and 3% Symsitive^®^ in butylene glycol completely inhibited nematocyst discharge and were considered inhibitor solutions (Figure 2C–E and Table 2).

#### 2.2.2. Test 3—Venom Load

After the identification of inhibitory solutions with TSA, venom activity assays were conducted using live tentacles in ex vivo TSBAA with only the reducer and inhibitor solutions (Table 2). Seawater, a neutral solution (Table 2) that was used as a control, obtained a hemolytic area of 60.77 ± 31.38% (Figure 3 and Figure 4A). All the solutions identified as inhibitors in the previous assay (Section 2.2.1) decreased the hemolytic area after the sting process. The smallest hemolytic areas were obtained after incubation with 10% lidocaine in ethanol (3.30 ± 7.19%), 1.5% hydroxyacetophenone in distilled water + butylene glycol (2.48 ± 4.54%), and 3% Symsitive^®^ in butylene glycol (11.00 ± 17.66%) (Figure 3 and Figure 4B,D,E). Butylene glycol obtained a value of 26.54 ± 19.64% for the hemolytic area (Figure 3 and Figure 4C). Significant differences were found between seawater (control) and all the compounds *(*** p* ≤ 0.001) (ANOVA test) (Figure 3). Statistical data can be consulted in Appendix A.

## 3. Discussion

Jellyfish stings represent a human health hazard and have a negative impact on the tourism sector [9,10,18,24]. In the Mediterranean region, *P. noctiluca* is responsible for a high number of incidents among beachgoers [10,18,22,24]. Its high toxicity along with its wide distribution and high frequency in the basin [15,21,22] have positioned *P. noctiluca* as a target species for preventive measures, especially those that help to prevent and mitigate the effects of its sting [23,28,34,40,41].

In the past, a wide variety of compounds have been tested for their effectiveness in treating scyphozoan stings (Table 3). Among the most deeply rooted traditional remedies are urine, urea and ammonia solutions [9,18]. Previous preclinical tests showed inhibition of *P. noctiluca* nematocyst discharge in a 20% ammonia solution [40], yet the immediate and massive nematocyst discharge observed herein questions the inhibitory potential of both ammonia and scented ammonia (Figure 1B,L and Table 1). Probably, these opposite results, as in the case of vinegar in Ballesteros et al. (2021) [28], are due to the use of different methodologies (Table 3). While the stimulation in Morabito et al. (2014) [40] was chemical-mechanical, with 20 min of incubation in a chemosensitizer compound (e.g., glutamate) followed by mechanical stimulation with gelatine-coated test probes, here we evaluated discharge using the TSA method, a widely-used technique in cnidarians to discern between activator and potential inhibitory compounds [28,29,30,36,37,42,43]. In accordance with our results, high nematocyst discharge and exacerbation of pain intensity were also identified during the ammonia test for the scyphomedusa *Chrysaora quinquecirrha* [37,39]. Despite promoting nematocyst discharge (Table 1), ammonia-based post-sting products expressly for treating *P. noctiluca* stings [41] are currently available on the market. We conclude that ammonia is not an inhibitor solution for *P. noctiluca* nematocysts and its use is contraindicated for the treatment of their stings. Regarding urine, in accordance with Pyo et al. (2016) [36] to *Nemopilema nomurai*, no nematocyst discharge was identified for *P. noctiluca* (Table 1), yet these results are in contrast to its effect for *Cyanea capillata* (Table 3).

Like ammonia solutions, vinegar or acetic acid are perceived by the public as beneficial products for the treatment of jellyfish stings. However, the stimulation of discharge and their non-inhibitory effect are well-documented for scyphozoans, including *P. noctiluca* [28,36,37,38,39]. Both here and in previous studies [37,44], 5% acetic acid has been used to chemically stimulate nematocyst discharge in scyphozoans (Table 2). Other acid solutions such as carbonated cola and lemon juice also triggered discharge (Figure 1J,K and Table 1), supporting the correlation between nematocyst discharge and extreme acidic solutions reported previously [45]. Despite the solid scientific evidence demonstrating its contraindication in scyphozoans (Figure 1M) [28,36,37,38,39,46], some administrations recommend the use of vinegar. For example, in the Balearic Islands, one of the major tourist destinations in Europe [47] and with a high presence of scyphozoans, the use of vinegar is still recommended to treat jellyfish stings [48]. A survey in Mallorca (Balearic Islands) [47] showed the added value of the presence of health services on beaches, but the effectiveness of such preventive measures can be diminished if the health service does not receive appropriate guidelines. In addition, first-aid protocols must be updated as research on the topic advances.

Solutions and slurry of sodium bicarbonate were strongly recommended in first-aid protocols for scyphozoan species, including those that inhabit the Mediterranean basin [39,49]. However, there is no robust scientific evidence to support this recommendation (Table 3). Morabito et al. (2014) [40] identified nematocyst discharge in *P. noctiluca*, which is in accordance with our results (Figure 1D,E and Table 1). Due to the nematocyst discharge observed herein (Table 1) and the lack of clinical evidence on topical relief of jellyfish sting symptoms (Table 3), the use of sodium bicarbonate has been removed from the Spanish guidelines on Mediterranean jellyfish stings in an updated version of the first-aid protocols [16].

Anionic solutions have been observed to promote nematocyst discharge in *P. noctiluca* [45,50]. While anions such as I^−^ and Cl^−^ triggered high rates of discharge, cations such as Mg^+^ and Ba^+^ did not elicit nematocyst discharge. Yet, when they were added to iodine solutions discharge was inhibited [45,50]. Salleo et al. (1948) [45] reported nematocyst discharge in sodium chloride in accordance with our results (Figure 1F and Table 1), but conflicting results have been obtained for choline chloride, barium choline and magnesium chloride (Table 3). Solutions containing lanthanum and gadolinium did not inhibit nematocyst discharge in this study as previously reported for *P. noctiluca* [51] (Table 2) unlike lidocaine solution (ANOVA, *p* ≤ 0.001) (Figure 2, Figure 3 and Figure 4B and Table 2) also formerly described for scyphozoans (Table 3).

Nematocyst discharge promoted by osmotic change after fresh water application is a premise in first-aid protocols, which is why it is not recommended [52,53]. Fresh water did not produce nematocyst discharge in *P. noctiluca* (Table 1) and it has been considered a non-activator solution in scyphozoans [36]. Distilled water is commonly used to induce the detachment of *P. noctiluca* cnidocytes from the epidermis, maintaining their integrity [45,54]. However, despite its neutral effect on the nematocyst discharge of *P. noctiluca* (Table 1), the use of fresh, tap, deionized or distilled water is not recommended due to their potential to isolate undischarged cnidocytes, since these can roll on the skin and be mechanically activated, inoculating a second venom load [28,29,42,43].

Glycols, such as butylene glycol, are used in a variety of cosmetic products [55]. Here, no discharge was observed with butylene glycol (Table 1), allowing us to validate the safe use of butylene glycol in preventive cosmetic products, in terms of non-activation of discharge, unlike products with ammonia [41] and acetic acid [56]. Moreover, butylene glycol was classified as a reducer solution (Table 2), and a decrease in hemolytic zones was later observed in TSBAA tests (ANOVA, *p* ≤ 0.001) (Figure 2B, Figure 3 and Figure 4C). Probably, due to its viscous consistency, 100% butylene glycol wraps around the tentacle to act as a physical barrier. Lower values compared to the other formulations in TSBAA (Figure 3) are possibly due to the loss of effectiveness of the physical barrier during the sting process. When butylene glycol was dissolved in water (e.g., 50% butylene glycol in distilled water, Table 2), its inhibitor effect was lost.

Jellyfish stings cause clinical signs such as erythema, redness or papules accompanied by pain, burning sensation and itching [14,23,24]. Active ingredients that reduce skin damage and/or pain or burning sensation, as well as inhibiting nematocyst discharge provide added value in preventive products. Here, the formulation containing Symsitive^®^ inhibited nematocyst discharge and significantly reduced hemolytic area (ANOVA, *p* ≤ 0.001) (Figure 2E, Figure 3 and Figure 4E and Table 2). In the TSBAA test, high variability between replicates was observed for Symsitive^®^ (Figure 3). Discharge of nematocysts could have occurred in some areas where the product has not fully penetrated. Symsitive^®^ is composed of trans-4-t-butylcyclohexanol, a TRPV1 receptor antagonist [57], and pentylene glycol [55]. In the cosmetic industry, this active ingredient has been clinically proven to relieve symptoms such as erythema, stinging and burning [58,59]. An inhibitor effect was also reported for 1.5% hydroxyacetophenone in distilled water + butylene glycol (ANOVA, *p* ≤ 0.001) (Figure 2, Figure 3 and Figure 4D and Table 2), an anti-irritant used in cosmetic products [55]. Although hemolysis was not observed in most replicates of 1.5% hydroxyacetophenone in distilled water + butylene glycol, some small hemolytic areas were observed in some replicates (Figure 3 and Figure 4D). The same occurred for lidocaine (Figure 3 and Figure 4B). This fact could be due to the fact that the compounds did not penetrate in some areas and some discharge of nematocysts could have occurred promoted by the weight (Section 4.4.2).

For the time being, the role of Symsitive^®^ and hydroxyacetophenone in the modulation of *P. noctiluca* nematocyst discharge is unknown, and further research is needed to determine these. However, due to their inhibitor effect and decrease in hemolytic areas (ANOVA, *p* ≤ 0.001), both Symsitive^®^ and hydroxyacetophenone are considered compounds of great value for the cosmetic industry and identified as active ingredients suitable for incorporation into sunscreens and cosmetic products against jellyfish stings.

**Table 3 marinedrugs-20-00571-t003:** Summary of the efficacy of different compounds for the class Scyphozoa.

Scyphozoan	Methodology/Metric	Compounds	Effect	Reference
*Pelagia* *noctiluca*	Tentacle Solution Assay/ Nematocyst discharge	Ammonia (10%), barium chloride (10%), bleach, scented ammonia, lemon juice	High discharge	Present study
		Sodium bicarbonate solutions (10%), sodium chloride (10%), papain (10%), acetic acid (5%), vinegar	Mild discharge	
		Seawater, bromelain (10%), choline chloride (10%), copper gluconate (10%), gadolinium (III) chloride hexahydrate (10%), iodine (10%), lanthanum (III) chloride hexahydrate (10%), magnesium chloride (10%), magnesium sulfate (10%), distilled water, fresh water, physiological saline, urine, butylene glycol (50%)	Neutral (not inhibitory)	
		Butylene glycol	Reducer (only some isolated nematocysts discharged)	
		Hydroxyacetophenone (1.5%) in distilled water + butylene glycol (1:1), lidocaine (10%) and 3% Symsitive^®^ in butylene glycol	Discharge inhibited	
	Tentacle Skin Blood Agarose Assay/Venom activity (hemolytic effect)	Butylene glycol, lidocaine (10%), butylene glycol, hydroxyacetophenone (1.5%) in distilled water + butylene glycol (1:1) and 3% Symsitive^®^ in butylene glycol	Decreased hemolysis	
	Chemical-mechanical stimulation/Nematocyst discharge	Sodium bicarbonate (10%)	Discharge	[40]
		Lidocaine (1%), ammonia (20%), ethanol (70%), acetic acid (5%)	Discharge inhibited	
	Tentacle Solution Assay/ Nematocyst discharge	Vinegar	Mild discharge	[28]
		Sea water	Neutral (not inhibitory)	
	Tentacle Skin Blood Agarose Assay/Venom activity (hemolytic effect)	Seawater, vinegar	No decrease in hemolysis	
	Cytotoxicity assays/Venom activity (cytolytic effect)	*Ananas comosus, Carica papaya*	Improved cell survival	[34]
	Tentacle solution assay/Nematocyst discharge	Anions (I^−^, Cl^−^, F^−^), choline chloride, potassium chloride, sodium chloride, lithium chloride, cesium chloride, potassium iodine, sodium iodine, potassium sulfate, sodium sulfate, ammonium sulfate	Discharge	[45,50]
		Calcium chloride, barium chloride, magnesium chloride	Discharge inhibited	
	Case reports/Pain, redness and edema	Jellywash^®^	Prevention or improvement of pain, redness and edema	[23]
*Cyanea* *capillata*	Tentacle solution assay/Nematocyst discharge)	Vinegar	Partial discharge	[43]
		Urine, isopropanol	Moderate discharge	
		Seawater	No discharge	
	Tentacle Skin Blood Agarose Assay/Venom activity (hemolytic effect)	Seawater, urine	Increased hemolysis	
		Vinegar, Sting No More^®^ spray	Decreased hemolysis	
	Randomized controlled trials/Pain and skin manifestations (color and structural changes and vesicles)	Safe Sea^®^ (sunscreen with prophylaxis)	Reduction in the number of subjects with pain, discomfort and skin manifestations	[32]
	Tentacle solution assay/Nematocyst discharge	Acetic acid (5%)	Discharge	[38]
		Methylated spirits	No discharge	
*Nemopilema* *nomurai*	Tentacle Solution Assay/ Nematocyst discharge	Acetic acid (4%), isopropanol	High discharge	[36]
		Distilled water, ethanol (70%), ethanol (20%)	Low discharge	
		Seawater, lidocaine (10%)	No discharge	
	Nonrandomized controlled trials/Pain, redness and erythema	Seawater, lidocaine (10%)	Relief of pain and redness	
		Acetic acid (4%), ethanol (70%), ethanol (20%), isopropanol	Increased pain and redness, erythema	
	Cytotoxicity assays/Venom activity (cytolytic effect)	Tetracycline	Inhibition of the cytotoxic effect	[60]
	Dermal toxicity test	Tetracycline and lanoline + tetracycline	Decreased the level of hemorrhage	
*Chrysaora* *quinquecirrha*	Tentacle Solution Assay/Nematocyst discharge	Ethanol (70%), ammonia (20%), bromelain (10%)	High discharge	[37]
		Lidocaine (4%)	Discharge inhibited	
		Acetic acid (5%)	Mild discharge	
		Seawater, urea (10%)	No discharge	
	Nonrandomized controlled trials/Pain, redness and erythema	Seawater, deionizer water, bromelain (10%)	No change in pain intensity	
		Lidocaine (5%)	Noticeable alleviation of pain	
		Lidocaine (10%)	Further reduction in pain	
		Lidocaine (15%)	Maximum reduction in pain	
		Ammonia (20%), acetic acid (5%), ethanol (70%)	Exacerbation of pain	
	Tentacle Solution Assay/Nematocyst discharge	Sodium hypochlorite, acetone, vinegar (acetic acid 5%), ammonia, magnesium chloride	High discharge	[39]
		Papain, baking soda slurry, Stingose^®^ (20% aluminum sulfate in detergent)	Discharge inhibited	
*Chrysaora* *fuscescens*	Randomized controlled trials/Pain, discomfort, erythema and edema	Safe Sea^®^ (sunscreen with prophylaxis)	Prevention and reduction in pain and erythema	[33]
*Chrysaora* *chinensis*	Tentacle Solution Assay/ Nematocyst discharge	Seawater, sodium bicarbonate, papain, lidocaine	No discharge	[46]
		Acetic acid, isopropylalcohol	High discharge	
	Randomized controlled trials/Pain, erythema	Papain	Decreased pain and erythema	
		Sodium bicarbonate	Decreased erythema	

## 4. Materials and Methods

### 4.1. Jellyfish Cultures

Jellyfish were cultivated in Loro Parque Aquarium (Tenerife, Spain). *P. noctiluca* adult individuals were grown for 80 days in kreisel tanks with continuous water renewal, 33–35‰ salinity, 22 °C temperature and light cycle of 12 h light/12 h dark. Individuals were fed daily with *Artemia* sp. nauplii, small pieces of the jellyfish *Aurelia* sp., eggs of *Merluccius merluccius* and *Acetes* sp. (white prawns).

### 4.2. Compounds

A total of 31 substances and formulations, including some compounds previously evaluated in scyphozoans (Table 3) and other new ones proposed, were tested. Ammonia, barium chloride, sodium chloride, acetic acid, ethanol, papain, bromelain, sodium chloride, choline chloride, copper gluconate, gadolinium (III) chloride hexahydrate, lanthanum (III) chloride hexahydrate, magnesium chloride hexahydrate, lidocaine and hydroxyacetophenone were purchased from Merck Sigma Aldrich. Symsitive^®^ was obtained from the Symrise company. Butylene glycol was purchased from the KH Neochem Co. Ltd. Urine samples were obtained from three volunteers. The rest of the compounds were easily-obtained everyday products.

### 4.3. Screening of Solutions

#### Test 1: Nematocyst Discharge—Tentacle Solution Assay (TSA)

Method 1. To evaluate possible nematocyst discharge, tentacle pieces approximately 3 cm long were incubated for 5 min. in microwells with 2 mL of each solution. Then, each sample was placed on a slide (76 × 26 mm) and observed under a light microscope to evaluate the nematocyst response.

The nematocyst response was classified qualitatively into four categories in accordance with Pyo et al. (2016) [36]:0: no discharge was observed;+: low discharge of nematocysts;++: medium discharge of nematocysts;+++: high discharge of nematocysts.

The effect of the rinse solution was classified into one of two categories:Activator effect solution: nematocysts were activated after incubation with the solution;Neutral effect solution: nematocysts were not activated after incubation with the solution.

### 4.4. Evaluation of Inhibitor Effect

#### 4.4.1. Test 2: Nematocyst Discharge—Tentacle Solution Assay (TSA)

After the screening of substances, activator solutions–those producing nematocyst discharge on incubation (Method 1)–were discarded. Only neutral solutions were evaluated.

Method 2. Tentacle pieces approximately 3 cm long were incubated in microwells with 2 mL of each neutral solution (Section 4.3) for 5 min. Then, they were placed on a slide (76 × 26 mm). In order to determine the inhibitor effect, tentacles were chemically stimulated by the application of 15 µL of 5% acetic acid solution (an activator solution *per se*) with a micropipette.

The nematocyst response was classified qualitatively into four categories [36]:0: no discharge was observed;+: low discharge of nematocysts;++: medium discharge of nematocysts;+++: high discharge of nematocysts.

The effect was classified into one of three categories:Neutral effect: nematocysts were not activated after the first incubation with the solution but did produce discharge with the consecutive chemical stimulation of 5% acetic acid solution;Reducer effect: nematocysts were not activated after the first incubation with the solution but isolated nematocyst discharge was observed with the subsequent chemical stimulation of 5% acetic acid solution in some areas;Inhibitor effect: nematocysts were not activated after the first incubation with the solution, nor after the consecutive chemical stimulation of 5% acetic acid solution.

#### 4.4.2. Test 3: Venom Load—Tentacle Skin Blood Agarose Assay (TSBAA)

Method 3. To determine the effect on the venom load (hemolytic area) of the reducer and inhibitor solutions identified during Method 2 (Section 4.4.1), we used an adapted protocol from the tentacle skin blood agarose assay (TSBAA) [30]. Briefly, an agarose gel preparation incorporating sheep red blood cells (SRBC) (Thermo Fisher Scientific) was used, covered by a thin tissue of pig intestine to simulate the effect of the human skin barrier [28].

After the preparation of the SRBC agarose rectangles and the pig small intestine, plastic molds to control the sting area were placed on top of the pig intestine (sting diameter of 57.35 mm^2^) [61]. Venom activity from captive individuals (Section 4.1) was previously validated in Ballesteros et al. (2022) [61].

Tentacle pieces (approx. 3 cm long) of *P. noctiluca* from aquaculture were incubated in each reducer and inhibitor solution (Section 4.4.1) for 5 min. Subsequently, tentacles were placed on the plastic molds [61] with a weight of 0.66 g to ensure contact. After 1 min of contact, the plastic molds and the intestine sections were removed and the SRBC agarose rectangles were stored in the humidification chamber at room temperature. After 22 h, photographs were taken of the hemolytic areas which were then calculated using the Fiji version of ImageJ software [62]. The data were tested for normality and homogeneity using the *stats* package available as part of the R basic software platform [63]. Subsequently, an ANOVA test was performed using the *aov* function to test significant differences between hemolytic areas, with an additional pairwise comparison using the *pairwise* function to test between which groups the differences occurred. Finally, a graphical representation was performed using the *ggplot2* package from the R software platform [64].

## 5. Conclusions

*Pelagia noctiluca* is a highly toxic jellyfish responsible for the majority of stings in Mediterranean waters, so it is considered a target species for preventive measures. Unlike lidocaine, which inhibited nematocyst discharge, solutions containing lanthanum and gadolinium were considered neutral solutions. The use of ammonia, vinegar, 5% acetic acid and baking soda, among others, is not recommended to treat *P. noctiluca* stings because they promote nematocyst discharge. Hydroxyacetophenone and Symsitive^®^ were identified as nematocyst inhibitor compounds of great value for the cosmetic industry. These active ingredients can be incorporated into sunscreens to reduce the symptoms of jellyfish stings, as well as in a rinse solution to help to remove tissue or/and residual cnidocytes after jellyfish stings.

## Figures and Tables

**Figure 1 marinedrugs-20-00571-f001:**
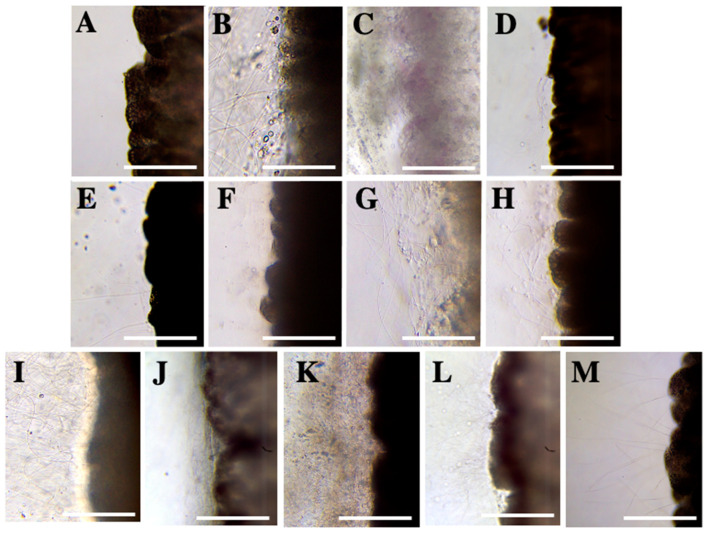
Screening of activator solutions: (**A**) Seawater, a neutral solution used as a control, (**B**) ammonia, (**C**) barium chloride, (**D**) sodium bicarbonate in seawater, (**E**) sodium bicarbonate in distilled water, (**F**) sodium chloride, (**G**) papain, (**H**) acetic acid, (**I**) bleach, (**J**) carbonated cola, (**K**) lemon juice, (**L**) scented ammonia, and (**M**) vinegar. Note the discharged tubules with activator solutions (**B**–**M**) compared with seawater, a neutral solution (**A**). Scale bars: 0.5 mm.

**Figure 2 marinedrugs-20-00571-f002:**
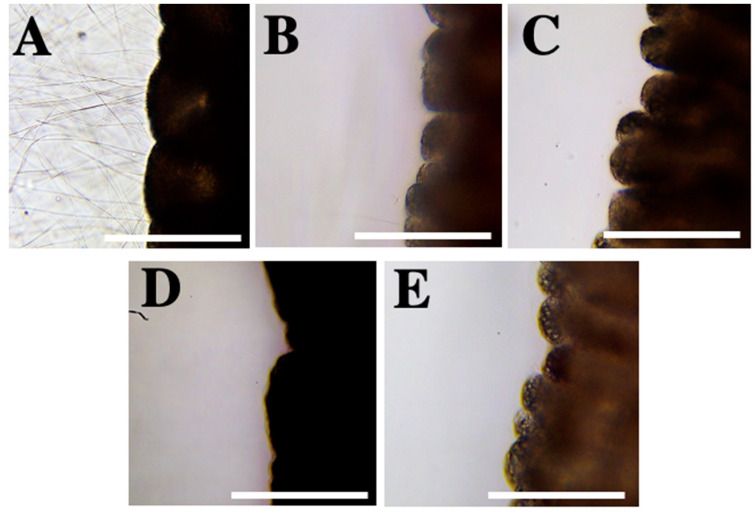
Inhibitor solutions identified after chemical stimulation with 5% acetic acid (test 2): (**A**) Acetic acid (chemical stimulator). Note all tubules discharged after its application; (**B**) Butylene glycol, a reducer compound; (**C**) 1.5% hydroxyacetophenone in distilled water + butylene glycol (1:1); (**D**) 10% lidocaine in ethanol and (**E**) 3% Symsitive^®^ in butylene glycol, inhibitory solutions. Scale bars: 0.5 mm.

**Figure 3 marinedrugs-20-00571-f003:**
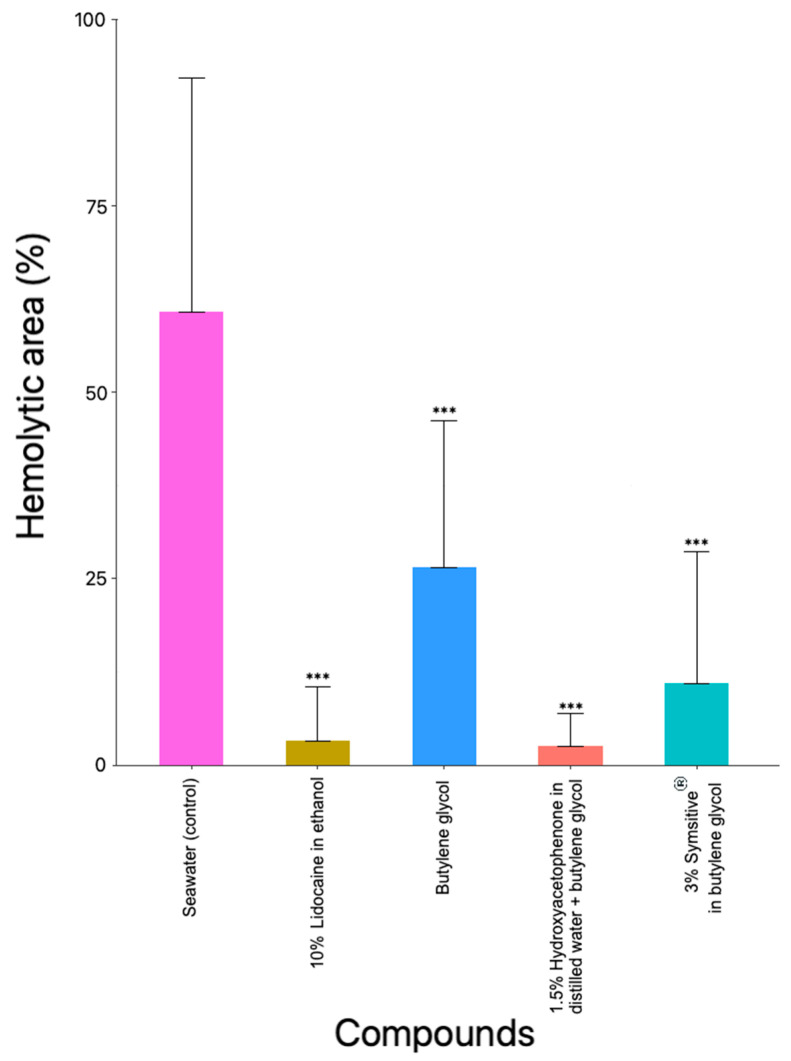
Hemolytic area (%) produced by the venom of *Pelagia noctiluca* in the presence of the inhibitor and reducer substances after 22 h on SRBC agarose. Significant differences were found between seawater (control) and the rest of compounds *(**** *p* ≤ 0.001) (ANOVA test). The number of replicates was 18 for seawater (control) and 16 for each compound.

**Figure 4 marinedrugs-20-00571-f004:**
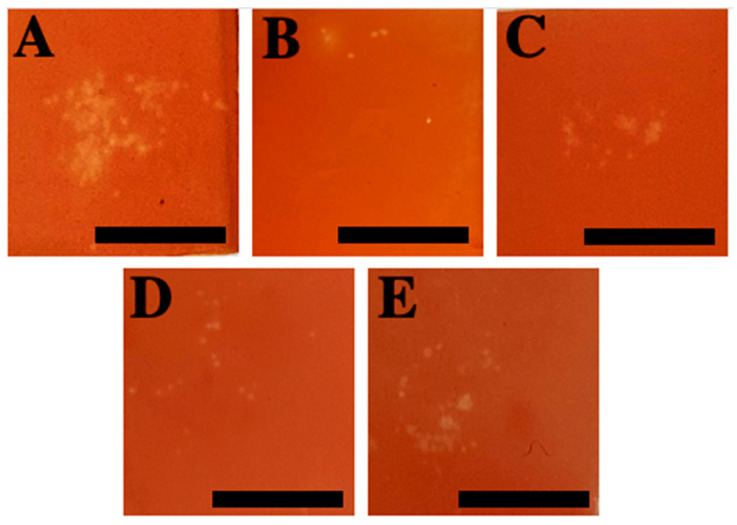
Hemolytic areas produced after 22 h of the sting of *Pelagia noctiluca* using the TSBAA method in: (**A**) seawater (control), (**B**) lidocaine in ethanol, (**C**) butylene glycol, (**D**) 1.5% hydroxyacetophenone in distilled water + butylene glycol (1:1) and (**E**) 3% Symsitive^®^ in butylene glycol. Scale bars: 10 mm.

**Table 1 marinedrugs-20-00571-t001:** Screening of potential solutions to inhibit nematocyst discharge in *Pelagia noctiluca*. Effect on nematocysts after incubation in each solution for 5 min (test 1).

Compounds	Test 1: Incubation		
	*n*	Discharge ^1^	Effect ^2^
Seawater (control)	3	0	Neutral
10% Ammonia in distilled water	8	+++	Activator
10% Barium chloride in distilled water	3	+++	Activator
Bleach	3	+++	Activator
Lemon juice	3	+++	Activator
Scented ammonia	3	+++	Activator
10% Sodium bicarbonate in seawater	3	++	Activator
10% Sodium bicarbonate in distilled water	3	++	Activator
10% Sodium chloride in distilled water	3	++	Activator
10% Papain in distilled water	3	++	Activator
5% Acetic acid in distilled water	3	++	Activator
Carbonated cola	3	++	Activator
Vinegar	3	++	Activator
10% Bromelain in distilled water	3	0	Neutral
10% Choline chloride in distilled water	3	0	Neutral
10% Copper gluconate in distilled water	3	0	Neutral
10% Gadolinium (III) chloride hexahydrate in distilled water	3	0	Neutral
10% Iodine in distilled water	3	0	Neutral
10% Iodine in seawater	3	0	Neutral
10% Lanthanum (III) chloride hexahydrate in distilled water	3	0	Neutral
10% Magnesium chloride hexahydrate in distilled water	3	0	Neutral
10% Magnesium sulfate in distilled water	3	0	Neutral
Distilled water	3	0	Neutral
Fresh water	3	0	Neutral
Physiological saline	3	0	Neutral
Urine	3	0	Neutral
10% Lidocaine in ethanol	3	0	Neutral
Butylene glycol	6	0	Neutral
Butylene glycol + distilled water (1:1)	7	0	Neutral
1.5% Hydroxyacetophenone in distilled water + butylene glycol (1:1)	6	0	Neutral
3% Symsitive^®^ in butylene glycol	8	0	Neutral

Method 1. Tentacle solution assay (TSA). ^1^ Nematocyst discharge categories: 0 = no discharge; ++ = medium discharge; +++ = high discharge. ^2^ Rinse solution effect categories: activator solution effect = nematocysts activated after incubation with the solution; neutral solution effect = nematocysts not activated after incubation with the solution. *n* indicates the number of replicates.

**Table 2 marinedrugs-20-00571-t002:** Inhibitory response of nematocyst discharge in *Pelagia noctiluca* after chemical stimulation with 5% acetic acid (test 2).

Compounds	Test 2: Discharge		
	*n*	Discharge ^1^	Effect ^2^
Seawater (control)	3	+++	Neutral
10% Bromelain in distilled water	3	+++	Neutral
10% Choline chloride in distilled water	3	+++	Neutral
10% Copper gluconate in distilled water	3	+++	Neutral
10% Gadolinium (III) chloride hexahydrate in distilled water	3	+++	Neutral
10% Iodine in distilled water	3	+++	Neutral
10% Iodine in seawater	3	+++	Neutral
10% Lanthanum (III) chloride hexahydrate in distilled water	3	+++	Neutral
10% Magnesium chloride in distilled water	3	+++	Neutral
10% Magnesium sulfate in distilled water	3	+++	Neutral
Distilled water	3	+++	Neutral
Fresh water	3	+++	Neutral
Physiological saline	3	+++	Neutral
Urine	3	+++	Neutral
Butylene glycol + distilled water (1:1)	6	+++	Neutral
Butylene glycol	6	+	Reducer
1.5% Hydroxyacetophenone in distilled water + butylene glycol (1:1)	6	0	Inhibitor
3% Symsitive^®^ in butylene glycol	8	0	Inhibitor
10% Lidocaine in ethanol	3	0	Inhibitor

Method 2. Inhibition of nematocyst discharge—Tentacle solution assay (TSA). ^1^ Nematocyst discharged categories: 0 = no discharge; + = low discharge; +++ = high discharge. ^2^ Rinse solution effect categories: Neutral = nematocysts were not activated after the first incubation with the solution but were activated by the consecutive chemical stimulation with 5% acetic acid; Reducer = nematocysts were not activated after the first incubation with the solution but isolated nematocysts were discharged after the chemical stimulation with 5% acetic acid in some areas; Inhibitor = nematocysts were not activated after the first incubation with the solution nor by the chemical stimulation with 5% acetic acid. *n* indicates the number of replicates.

## Data Availability

Not applicable.

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
