# Peer review of "Inhibition of Nematocyst Discharge from Pelagia noctiluca (Cnidaria: Scyphozoa)—Prevention Measures against Jellyfish Stings"

_marinedrugs, 2022, doi:10.3390/md20090571_

Round 1

Reviewer 1 Report

This manuscript is very well written because it is written logically and easily understood. The authors reviewed the previous studies on the inhibition of nematocyst release and explored new potential inhibitors. A variety of common substances were selected for the experiment, and several reagents that had not been tested before were explored for the experiment, and positive results were obtained. 1.5% hydroxyacetophenone, 3% Symsitive and butylene glycol are novel substances proposed in this manuscript that inhibit the emission of jellyfish nematocysts and have the potential to be used in cosmetics to prevent jellyfish stings.

There are some problems, which should be adjusted before it is considered for publication including discussion and data aspect. First question of this paper is the lack of full interpretation of experimental material selection, the author discussed in part to the existing of different material nematocyst release of stimulation and restraint effect had a good discussion, but there is no why to choose several  material s have inhibition effect, so the authors need to explain the cause of the material choice. Another problem in this manuscript is the experimental method. In the experiment on hemolytic area (TSBAA), the tentacles were placed in the inhibition solution for 5 min and then placed on the evaluation model, which is an improvement over the previous experiment. The nematocysts were fired through the pig intestinal barrier and the toxin was injected or spread on the agarose side. In this experiment, the hemolysis area had to wait for 22h after tentacle removal to be photographed. So the question is, why did it take so long to photograph? Is the initial phenomenon not obvious or difficult to photograph with the camera? Is the slow hemolysis due to agarose? The experimental part needs to be discussed.  And the photos of hemolysis should be added.

There are two small problems in the data of this manuscript. The first problem is that the figures and tables cannot be used at the same time, and the hemolysis area data appear in both tables and figures. Problem two appears in the table, the inhibition rate of hemolysis area is not marked with standard deviation.

In the conclusion part, in addition to the outstanding contribution of the discovery, the author briefly evaluated the role of traditional compounds in regulating the discharge of jellyfish nematocysts. Some results of Method 2. were not summarized, for example, some neutral substances which used to be consider as inhibitory solution showed more release after treatment with 5% acetic acid.

Author Response

Response to Reviewer 1 Comments

There are some problems, which should be adjusted before it is considered for publication including discussion and data aspect. First question of this paper is the lack of full interpretation of experimental material selection, the author discussed in part to the existing of different material nematocyst release of stimulation and restraint effect had a good discussion, but there is no why to choose several materials have inhibition effect, so the authors need to explain the cause of the material choice.

We evaluated the substances that had been previously tested (some for other cnidarians) with the aim of reevaluating them to search for possible inhibitors. In addition, we tested some substances that were commonly used in cosmetics to assess whether they were activator/neutral/inhibiting. Noting that two of them were inhibitory, they have been the subject of a patent. Now, with the results obtained, new studies with the objective of know the mechanism(s) of action should be carried out.

We had added “A total of 34 substances and formulations, including some compounds previously evaluated in scyphozoans (Table 3) and other new ones proposed, were tested”. in materials and methods (Line 275). Also, we have added more information about method 1 and 2.

Another problem in this manuscript is the experimental method. In the experiment on hemolytic area (TSBAA), the tentacles were placed in the inhibition solution for 5 min and then placed on the evaluation model, which is an improvement over the previous experiment. The nematocysts were fired through the pig intestinal barrier and the toxin was injected or spread on the agarose side. In this experiment, the hemolysis area had to wait for 22h after tentacle removal to be photographed. So, the question is, why did it take so long to photograph? Is the initial phenomenon not obvious or difficult to photograph with the camera? Is the slow hemolysis due to agarose? The experimental part needs to be discussed.  And the photos of hemolysis should be added.

Hemolysis forms slowly. The first hours appear white dots that expand as time goes by and, around 18, 22 or 24 hours can be photographed with a normal camera. It is not a problem of the agarose, but a matter of waiting until complete hemolysis is formed. Here we leave other articles that have used this methodology (and the authors of this methodology), so that you can observe the waiting time until the final hemolysis is completely formed.

We have added a new figure with the hemolytic areas (Figure 4).

  1. Wilcox, C.; Headlam, J.; Doyle, T.; Yanagihara, A. Assessing the efficacy of first-aid measures in Physalia envenomation, using solution- and blood agarose-based models. Toxins 2017, 9, 149, doi:10.3390/toxins9050149.
  2. Doyle, T.; Headlam, J.; Wilcox, C.; MacLoughlin, E.; Yanagihara, A. Evaluation of Cyanea capillata sting management protocols using ex vivo and in vitro envenomation models. Toxins 2017, 9, 215, doi:10.3390/toxins9070215.

There are two small problems in the data of this manuscript. The first problem is that the figures and tables cannot be used at the same time, and the hemolysis area data appear in both tables and figures. Problem two appears in the table, the inhibition rate of hemolysis area is not marked with standard deviation.

We have removed Table 3 and the inhibition percentages. They have also been removed from the rest of the text and materials and methods.

In the conclusion part, in addition to the outstanding contribution of the discovery, the author briefly evaluated the role of traditional compounds in regulating the discharge of jellyfish nematocysts. Some results of Method 2. we’re not summarized, for example, some neutral substances which used to be consider as inhibitory solution showed more release after treatment with 5% acetic acid. 

Added to the conclusion part.

Reviewer 2 Report

Authors have experimented with many chemicals for their study on the "Inhibition of Nematocyst Discharge from Pelagia noctiluca (Cnidaria: Scyphozoa)—Prevention Measures against Jellyfish Stings" from the Mediterranean Sea. 

Overall discussion is acceptable. Scyphozoans causing sting and scars, photos can authors publish here? I used to find many scars for cubozoans in Australia and other countries, but for scyphozoans not that much.  I have attached the marked pdf file for authors corrections. 

Italics need to be marked at some places and can you justify the mode of action of that particular chemical which inhibit the discharge of nematocysts. Even all chemicals mode of action is explained, it would be interesting. Authors can also mention the cases happened in this study. How they can compare with cubozoans stinging and scars? it can be discussed.

Can you write about its high toxicity and abundance? 

It must also be noted however, that acetic acid is recommended for most jellyfish stings due to the difficulty in identifying the species of jellyfish responsible. Although scyphozoans are the prevalent class in the Mediterranean, the cubozoan Carybdea marsupialis, which can cause severe systemic effects is also known to be present there (Bordehore et al., 2014). In such events, it is better to err on the side of caution and use acetic acid, as although scyphozoan stings are painful, they do not have as great a potential to cause severe systemic effects compared to cubozoans.  

Author Response

Dear reviewer 2, 

Please see the attachment. Thank you very much for your comments and contributions.

Round 2

Reviewer 1 Report

The smallest hemolytic areas were obtained after incubation with 10% lidocaine in ethanol (3.30 ± 7.19%), 1.5% hydroxyacetophenone in distilled water + butyl- 145 ene glycol (2.48 ± 4.54%), and 3% Symsitive in butylene glycol (11.00 ± 17.66%) .  SD of the hemolytic areas is  higher than the average value. Please explain the reasons in results or discussion.

Author Response

The smallest hemolytic areas were obtained after incubation with 10% lidocaine in ethanol (3.30 ± 7.19%), 1.5% hydroxyacetophenone in distilled water + butyl- 145 ene glycol (2.48 ± 4.54%), and 3% Symsitive in butylene glycol (11.00 ± 17.66%).

SD of the hemolytic areas is higher than the average value. Please explain the reasons in results or discussion.

The TSBAA shows a variability between replicates (also in other articles). To improve this, we control the sting area as the tentacle contracting and expanding gives rise (mainly to great variability) (Ballesteros et al., 2022). We also increased the number of replicates to 16 when other studies use 3 or 6. In addition, we also added a weight to favor, even more, the discharge of nematocysts. Furthermore, nematocyst batteries do not contain the same number of nematocysts, nor types.

In the case of the compounds that decreased, it was due to the fact that in some replicates (e.g. 1.5% HAP in water + Butylene Glycol) (4 replicates of 16) they presented small foci that could be due to the fact that the compounds did not penetrate into some areas not so superficial and the weight caused the discharge.

We have added this in the discussion (yellow).

Reviewer 2 Report

Pelagia noctiluca is a highly toxic jellyfish responsible for the majority of stings in Mediterranean waters. Since authors have written like this, I would like to see more information as like follows as a major revision before the acceptance.

Revise: L29: belonging to the Phylum Cnidaria.

Provide the pictures of Pelagia noctiluca.

Table 3: N. nomurai nomurai, why it is given 2 times - revise it.

Jellyfish were cultivated in Loro Parque Aquarium (Tenerife, Spain). P. noctiluca adult individuals were grown in kreisel tanks with continuous water renewal, 33–35‰ salinity, 22ºC temperature and light cycle of 12h light/12h dark. - It's interesting, can you provide the pictures of culture tank?

You have taken cultured adult for this study? - How old? write in detail? Did you observe the younger ones for this inhibition study, why not? How long it was cultured?

Feed Individuals were fed daily with Artemia sp. nauplii, small pieces of the jellyfish Aurelia sp., eggs of Merluccius merluccius and Acetes sp. (white prawns) - Are they play any role in the production of toxin in P. noctiluca? Discuss in detail.

Aurelia is not a mild toxic species?, if you feed Aurelia, will it enhance the toxicity level of P. noctiluca? Need to write in detail about the toxicity of Aurelia sp. If pictures are provided for the feed, would be more interesting.

L361-362 - incomplete??

The use of ammonia, vinegar, 5% acetic acid and baking soda, among  others, is not recommended to treat P. noctiluca stings because they ??

Line 355-362 is confusing, authors not read/written properly, it seems it is submitted in a hurry, not a right way to do for this prestigious journal. 

421-422 realign.

Why Gershwin papers on cubozoans toxicity is not well discussed here? Why vinegar is effective for cubozoans and not for scyphozoans, need a concrete explanation. 

Author Response

Thank you very much

Round 3

Reviewer 2 Report

It is improved well, better to thank several reviewers of this manuscript in acknowledgement.